# Simulating Metabolic Flexibility in Low Energy Expenditure Conditions Using Genome-Scale Metabolic Models

**DOI:** 10.3390/metabo11100695

**Published:** 2021-10-12

**Authors:** Andrea Cabbia, Peter A. J. Hilbers, Natal A. W. van Riel

**Affiliations:** 1Department of Biomedical Engineering, Computational Biology, Eindhoven University of Technology, Groene Loper 5, 5612 AE Eindhoven, The Netherlands; a.cabbia@tue.nl (A.C.); p.a.j.hilbers@tue.nl (P.A.J.H.); 2Amsterdam University Medical Centers, University of Amsterdam, Meibergdreef 9, 1105 AZ Amsterdam, The Netherlands

**Keywords:** metabolic flexibility, respiratory quotient, energy expenditure

## Abstract

Metabolic flexibility is the ability of an organism to adapt its energy source based on nutrient availability and energy requirements. In humans, this ability has been linked to cardio-metabolic health and healthy aging. Genome-scale metabolic models have been employed to simulate metabolic flexibility by computing the Respiratory Quotient (RQ), which is defined as the ratio of carbon dioxide produced to oxygen consumed, and varies between values of 0.7 for pure fat metabolism and 1.0 for pure carbohydrate metabolism. While the nutritional determinants of metabolic flexibility are known, the role of low energy expenditure and sedentary behavior in the development of metabolic inflexibility is less studied. In this study, we present a new description of metabolic flexibility in genome-scale metabolic models which accounts for energy expenditure, and we study the interactions between physical activity and nutrition in a set of patient-derived models of skeletal muscle metabolism in older adults. The simulations show that fuel choice is sensitive to ATP consumption rate in all models tested. The ability to adapt fuel utilization to energy demands is an intrinsic property of the metabolic network.

## 1. Introduction

The ability of an organism to efficiently switch between the oxidation of different energy substrates according to environmental circumstances is known as metabolic flexibility. Healthy metabolism is characterized by physiological shifts between glucose and fat oxidation in response to nutrient availability. This process maintains homeostasis in response to changing energy demands, for example, during exercise. This transition is driven by insulin activity and regulated by a cross-talk between metabolic and signaling pathways across different tissues [1]. Skeletal muscle, as the largest contributor to insulin-mediated glucose uptake from plasma and as a major determinant of energy expenditure in resting and non-resting conditions [2], is one of the major drivers of metabolic flexibility.

Energy metabolism is heavily involved in the aging process, not only because mitochondrial dysfunction and impaired nutrient sensing are among the main drivers of the aging process [3], but also because all the recognized hallmarks of aging are connected to undesirable metabolic alterations [4]. Metabolic flexibility is recognized as a feature of healthy metabolism and has been associated with longevity and a longer health span. It has also been associated with increased insulin sensitivity [5] and lower incidence of age-related diseases, such as type 2 diabetes [6] and cardiovascular diseases [7]. Treatments targeting metabolic flexibility may delay the onset of aging and related comorbidities. Currently, regular physical activity and a balanced diet are still the best available treatments to increase metabolic health and to maximize health span [8,9].

Computational models are key to investigate the complexity of the interactions between nutrition and physical activity. Constraint-based metabolic models have been successfully used to simulate metabolic flexibility in silico by computing the Respiratory Quotient (RQ) in different nutritional conditions, for example, after a meal, during the transition between the fast and the fed state [10,11]. The RQ value concerns which macro nutrients are metabolized and which pathway is used for energy production. It is defined as the ratio of carbon dioxide produced by the body to oxygen consumed by the body, and it varies between values of 0.7 for pure fat metabolism and 1.0 for pure carbohydrate metabolism.

While the influence of nutrition and diet composition on metabolic flexibility is well documented [12,13], fewer studies have examined the role of physical activity and sedentary behaviors on metabolic flexibility. Previous studies which modeled RQ during the fast to fed transition using constraint-based models did not take the effect of energy expenditure on RQ into consideration [11]. In this study, we propose a new description of the fast to fed transition that allows us to simulate the effect of various levels of physical activity on fuel choice in constraint-based models.

Constraint-based metabolic models do not include any description of signaling pathways. To simulate the changing concentration of plasma glucose and fatty acids after a meal, Nogiec and coworkers [11] directly modulated the fluxes through glucose and palmitate transporters, reactions transporting substrates between external medium and cytoplasm compartments. The maximization of ATP, creatine phosphate, glycogen, and triglycerides production was used as the objective function.

This implementation is not applicable to large genome-scale metabolic models such as Recon2.2 and Recon3D, which have multiple alternative transporters for glucose and palmitate coupled with the symport or antiport of different ions, such as H${}^{+}$ and Na${}^{+}$. In our model, we avoid this bias by limiting the availability of glucose and palmitate through exchange reactions to simulate the fast to fed transition. This is comparable to controlling the maximal amount of nutrients present in the external medium of a cell culture. ATP phosphodiester bond hydrolysis (ATPH) was chosen as the objective function. By maximizing ATP consumption instead of ATP production, we let the models generate ATP using the optimal pathway, thus eliminating another potential source of bias. By constraining the flux through the ATPH reaction, we can simulate a condition of reduced energy expenditure. A simplified visualization of the two models is presented in Figure 1.

In this study, we investigate the link between physical inactivity and metabolic flexibility by simulating the effect of changing levels of energy expenditure on fuel choice, measured as RQ. Our new description of metabolic flexibility is tested in a set of constraint-based metabolic models. This model set includes two human metabolic reconstructions, Recon2.2 [14] and Recon3D [15], a model of central carbon metabolism, MitoCore [16] and a set of 24 patient-derived models of skeletal muscle metabolism [17]. We show that, in all models tested, fuel choice is sensitive to ATP consumption rate, and that a reduction in ATP consumption reproduces phenotypes associated with metabolic inflexibility.

## 2. Results

### 2.1. New Description of the Fast to Fed Transition in Genome-Scale Metabolic Models Highlights Heterogeneity of Model Predictions

We tested our model of the fast to fed transition using three different constraint-based metabolic models—Recon2.2, Recon3D and MitoCore—to predict RQ in high energy expenditure conditions, meaning that the objective function ATPH was unconstrained (default upper bound ATPH = 1000 mM/gDw/h). Recon2.2, Recon3D and MitoCore share most of their reaction identifiers and were chosen to facilitate a comparative analysis of the results. To simulate the fasting condition, we restricted the maximal influx of glucose and palmitate to 0.5 mM/gDw/h and 0.38 mM/gDw/h, respectively. In the fed condition, the maximal influx of glucose and palmitate was restricted to 4.5 mM/gDw/h and 0.034 mM/gDw/h, respectively. The bounds were progressively changed to simulate the transition between these states. The exchange of other metabolites with the external medium was deactivated, except for the exchange of water and protons (H${}^{+}$).

All models predicted RQ values within the expected range (0.7 < RQ < 1.0), (Figure 2A), but the exact predictions by the three models were not consistent. In particular, the predictions of the MitoCore model were divergent from those of Recon2.2 and Recon3D. MitoCore’s RQ profile rose to an RQ value of 1.0 in the second half of the transition. An inspection of the uptake fluxes during the fast to fed transition showed that the model maximizes the uptake of glucose but does not metabolize palmitate in the second part of the simulation, despite its availability in the medium. This suggests that the composition of the medium employed in this study might be sub optimal to conduct metabolic flexibility simulations with the MitoCore model. Different RQ profiles can be explained by the different ATP yields for glucose and palmitate among the three models (Figure 2B). MitoCore has the highest ATP yield for glucose (33 mM/gDw/h), which explains why MitoCore selects glucose as its only energy substrate when sufficient glucose is available to meet energy requirements (glucose > 2.7 mM/gDw/h, Figure 2A). Recon2.2 and Recon3D have an ATP yield for glucose of 31.5 and 32 mM/gDw/h, respectively. The ATP yield for palmitate is 106.8 mM/gDw/h in Recon2.2, 113.0 mM/gDw/h in Recon3D and 111.9 mM/gDw/h in MitoCore. Moreover, the structure of the metabolic network differs significantly across the models. MitoCore is a comparably small model focused on mitochondrial metabolism, which only contains 555 reactions and only describes a part of the metabolic network of the Recon models. Additionally, the topology of the metabolic network of the MitoCore models is different from the topology of both Recon models due to a different formulation of mitochondrial transport reactions and of the proton gradient that drives oxidative phosphorylation, suggesting that the topology and the stoichiometry of the metabolic network also affect RQ predictions. These differences influence the type and the amount of substrate used by the model to fulfil the objective function, and thus determine the predicted RQ value. Both Recon3D and MitoCore models needed to be modified before they could predict RQ values in the expected range, as explained in Appendix B.

### 2.2. RQ Is Sensitive to Changes in ATP Consumption Rate

We showed that our model of the fast to fed transition could reproduce theoretical RQ values in high energy expenditure conditions. To investigate how each model adapts its fuel choice to a decrease in energy expenditure, we progressively reduced the rate of ATP consumption. Figure 3 shows the RQ profiles computed during the fast to fed transition for the Recon2.2, Recon3D and MitoCore models, while the upper bound on the ATPH reaction was decreased within the range of 1–200 mM/gDw/h. The RQ values were sensitive to changes in the ATP consumption rate in all models tested: as the upper bound of the ATPH reaction decreased, RQ converged to a value of 1.0 in both the fast and fed conditions, as shown in Figure 3, indicating that the models prioritized carbohydrates as energy substrates in low energy expenditure conditions. Reduced ΔRQ, defined as the difference between the RQ in the fast state and in the fed state, is a phenotype associated with metabolic inflexibility.

The tipping point for this metabolic inflexibility phenotype is reached when the ATP consumption constraint becomes lower than the ATP generated in the fed state. The precise energy consumption level below which metabolic inflexibility phenotype develops is determined by the ATP yield for glucose and palmitate, and therefore is different for each of the three models (Figure 2B). A second tipping point is reached when ATP consumption becomes lower than the ATP generated in the fast state. After this point, the models exclusively employ glucose for ATP production.

Standard flux balance analysis (FBA) simulations are known not to generate a single unique solution. We observed fluctuations in the predicted RQ value, as shown in Appendix A, which were caused by unbalanced H${}^{+}$ in transport reactions and by the consequent rise in unfeasible reaction cycles in the flux distribution. After the imposition of further constraints on transport reactions between the cytoplasm and external compartments and the application of parsimonious FBA (PFBA) [18], the predicted RQ returned to the expected range. The objective values (i.e., the flux through the ATPH reaction) achieved over the course of the PFBA simulations for each of the three models are shown in Appendix A.

Despite the differences observed in the predictions of the three models, we can also observe a common pattern in the response of each model. Predicted RQ value is sensitive to changes in the upper bound of the ATPH reaction, i.e., sensitive to levels of energy expenditure [19].

### 2.3. RQ Changes Are Independent from Intake Fluxes

Our findings show that fuel utilization is dependent on energy expenditure, i.e., RQ is dependent on the upper bound of the ATPH objective reaction. To demonstrate that RQ changes are not due to artificial constraints on intake fluxes, we kept the bounds for palmitate and glucose intake fluxes constant while progressively reducing the upper bound of the ATPH reaction. In this simulation, the upper bound for glucose uptake was kept at 4.5 mmol/gDw/h, while the upper bound on palmitate intake flux was kept at 0.38 mmol/gDw/h. This simulation was performed using parsimonious constraints (PFBA).

RQ change is independent of substrate intake fluxes (Figure 4). We inspected the PFBA solutions relative to the two extreme conditions (i.e., for ATPH UB = 100 and ATPH UB = 1) and visualized these flux distributions using the Minerva software [20]. Despite the constraints on boundary reactions, we could still detect unfeasible loops in the PFBA solution (Appendix A) and the presence of proton leaks between the cytoplasm and mitochondrial matrix.

### 2.4. Resistance Training Increases Metabolic Flexibility

We established the importance of ATP phosphodiester bond hydrolysis rate for fuel selection in a set of generic human metabolic reconstructions and we hypothesized that low energy expenditure could be one of the major contributors to the development of metabolic inflexibility. Now, we ask whether a physical activity intervention, such as a resistance training program, can restore metabolic flexibility. To answer this question, we used a set of patient-derived models of skeletal muscle metabolism in older adults [17]. These models were developed using longitudinal gene expression data collected from skeletal muscle of the same individuals before and after a resistance training program. Therefore, they capture the long-term metabolic adaptations in energy metabolism that follow a metabolic intervention, such as a 12-week training program, and they can be used to investigate the effect of a non-nutritional intervention on metabolic flexibility.

In high energy expenditure conditions (ATPH upper bound = 1000 mM/gDw/h), all 24 models predicted identical RQ values (Figure 4A), meaning that they used the same mixture of substrates to produce ATP. Conversely, when ATPH was constrained to simulate a low energy expenditure condition (ATPH upper bound = 35 mM/gDw/h), each individual model predicted a different RQ profile (Appendix A). In low energy expenditure conditions, much less ATP was needed to fulfill the cellular objective and the models could use different mixtures of substrates to generate ATP. Trained models predicted a lower average RQ (Figure 4B), and a higher utilization of OXPHOS for ATP production (Figure 5B) during the fasting state in low expenditure conditions (ATP UB = 35 mM/gDw/h).Three untrained models (id: 4, 5, 11) predicted no flux through the mitochondrial adenine nucleotide translocator (ANT) reaction (reaction ID: ATPtm); therefore, they were not included in Figure 5.

Higher RQ and lower flux through OXPHOS in fasting conditions are phenotypes associated with insulin resistance and metabolic inflexibility [21]. Resistance training has been proved to be effective in restoring mitochondrial function in insulin resistant and diabetic subjects [22,23]. The results of this simulation show that a 12-week training intervention was effective in increasing utilization of the OXPHOS pathway in the skeletal muscle of older individuals. The large variability in OXPHOS utilization, especially among the untrained models, suggests that these individuals could have had different metabolic health before the beginning of the training program, and that some of them could have been more metabolically flexible than others. Our previous study on the same set of patient-derived models arrived at similar conclusions [17]. Without supplementary information regarding the lifestyle of these individuals before and during the study, for example, data about their nutrition or previous fitness status, we cannot speculate further. Therefore, this observation underlines the importance of collecting information about the lifestyle of the participants along with molecular data in systems medicine studies. Taken together, these results not only confirm that patient-derived models developed from longitudinal gene expression data can capture long-term metabolic adaptations to lifestyle change interventions, but also support the hypothesis that energy expenditure is a main determinant of metabolic flexibility and that physical activity can improve metabolic health in older adults.

## 3. Discussion

Metabolic flexibility is an important integrative biology concept which can help us understand the link between sedentary behavior, overnutrition and the dysregulation of energy metabolism and is an important part of metabolic health. Knowledge of the determinants of metabolic flexibility will help develop treatments to maintain and restore metabolic health in pathologies associated with metabolic inflexibility, such as insulin resistance, Type 2 diabetes, cardiovascular diseases, and aging.

Previous computational models of metabolic flexibility focused on nutritional intake as a determinant of metabolic flexibility, while the effect of energy expenditure on fuel choice remained understudied. In this study, we propose a new description of metabolic flexibility in genome-scale metabolic models, which enables the study of the interactions between physical activity and nutrition.

Patterns in fuel oxidation are determined not only by dietary intake but also by energy expenditure. We constrained the flux through the ATPH to reproduce a condition of lower energy expenditure. Limiting the flux through this reaction had a large effect on RQ and was sufficient to reproduce phenotypes associated with metabolic inflexibility, such as a lower ΔRQ between the fast and fed states.

When the flux through the ATP phosphodiester bond hydrolysis (ATPH) reaction was progressively reduced, RQ values progressively increased, while the difference between RQ in the fast and fed condition decreased. Since low energy expenditure is one of the main determinants of metabolic inflexibility, a physical activity intervention should restore metabolic flexibility even in absence of a nutritional intervention. To verify this hypothesis, we simulated the fast to fed transition in a set of patient-derived models of skeletal muscle metabolism that describe the metabolism of 12 older individuals before and after a 12-week resistance training program [24]. In high energy expenditure conditions, all models had the same response during the fast to fed transition. In low energy expenditure conditions (ATPH upper bound = 35 mM/gDw/h), trained models showed an increased utilization of the oxidative phosphorylation (OXPHOS) pathway for energy production (Figure 6). These results show that patient-derived models can capture some of the long-term metabolic adaptations resulting from a metabolic intervention, supporting the idea that these models can be used to improve our understanding of individual responses to diet and exercise.

Constraint-based metabolic models are useful tools to investigate the interactions between physical activity and nutrition, and how they influence metabolic health and the aging process. Nevertheless, there are still important inconsistencies in the predictions of different models. FBA predictions are heavily biased by the reaction composition of the models and by the boundary constraints the modelers apply. This fact is often overlooked in genome-scale metabolic modeling studies. Unfortunately, this is necessary due to the characteristics of the genome-scale modeling framework itself, especially when dealing with larger human models such as Recon2.2 and Recon3D, which contain a large number of boundary reactions. Leaving these models under-constrained often causes the appearance of physically unfeasible and physiologically unrealistic flux cycles in the FBA solution. We applied constraints to boundary reactions and used methods such as parsimonious FBA (PFBA) to reduce this effect. The differences that can be observed in the results are also due to the fact that different models have different focuses: MitoCore is limited to central carbon metabolism, Recon models are “generic” metabolic reconstructions, and skeletal muscle models collectively represent a tissue-specific metabolism. It is important to remember that none of these models are “true”; they are different approximations of human physiology.

However, we can identify properties that are common to all models: RQ predictions are dependent on the upper bound of ATPH reactions, even when palmitate and glucose intake fluxes are kept constant. Moreover, RQ change is independent of endocrine (insulin) signaling and allosteric regulation, which cannot be described in GSMMs. Metabolic flexibility thus appears to be an intrinsic property of the metabolic network, which is determined not only by intake fluxes but also by energy expenditure and by the model-specific ATP yields of relevant substrates.

In principle, the calculation of RQ from reaction stoichiometry is straightforward. In practice, it is difficult to obtain reproducible results in RQ simulations across different metabolic models.

These simulations are challenging not only because of their sensitivity to ‘technical’ variability, for example, the use of a different solver software, or due to different model composition and constraints on intake fluxes, but also because they are sensitive to "biological" variability, for example, different nutrition and energy expenditure habits, or different genetic backgrounds among different individuals. It may be difficult to distinguish "technical" variability from "biological" variability. Several model inconsistencies that biased the results were addressed, as discussed in Appendix B. To ensure the reproducibility of the results, model composition and simulation parameters such as reaction bounds should be standardized as much as possible.

Metabolic flexibility is an important health concept that integrates nutrition and energy expenditure. Expanding this concept to any response of fuel metabolism to external stressors, such as hot and cold temperatures, traumatic events such as illness, injuries or surgeries, and psychological stress [25] could reveal more details about how these factors interact in many pathological and physiological conditions, including aging.

## 4. Materials and Methods

### 4.1. Development of Patient-Derived Models of Skeletal Muscle Metabolism in Older Adults

Patient-derived models of skeletal muscle metabolism used in this study were developed using gene expression data [24], in combination with a template human metabolic network reconstruction, Recon2.2 [14]. Gene expression data were used to identify model reactions with high experimental support. Gene expression data were converted into an “ experimental confidence” score, ranging from 3 (high confidence) to −1 (negative confidence) for each reaction in the Recon2.2 model. Microarray probeset intensity values were log-transformed and sorted by decreasing expression values. When two or more probes mapped to the same gene, their expression values were averaged. Using these data, together with the gene–protein reaction annotations included in the Recon2.2 model, we computed the reaction-level confidence score which was used as input for the model-building algorithm. The CORDA algorithm [26] generated models which included all the high-confidence reactions and a minimal amount of lower-confidence reactions, such that the resulting networks were fully connected and that all the reactions could carry flux. After all the patient-derived models were drafted, they underwent substantial manual curation and validation. Details on the development and validation of the patient-derived models used in this study are presented in [17].

### 4.2. Simulating the Fast to Fed Transition in Constraint-Based Models

In this study, we investigate the effects of physical activity on metabolic flexibility using a set of different models, including Recon2.2, Recon3D, MitoCore, and 24 patient-derived models of skeletal muscle metabolism based on Recon2.2. Recon models are known to have several shortcomings: for example, the involvement of FAD in beta oxidation and the electron transport chain are not correctly represented. Successive studies made substantial corrections to these models [27]. The theoretical ATP yield for 1 mM palmitate (C16:0) is 104 mM/gDw/h. Recon2.2 predicts a yield of 106.75 mM/gDw/h, while the two corrected model version presented in [27] predict a yield of 108.25 and 108.24 mM/gDw/h, respectively. Since the discrepancy between the theoretical value and the predicted yield increased after the corrections, we believe that these corrections are not relevant for the particular substrates used in our simulations. For this reason, we decided to employ the “base” Recon2.2 model in this study. In addition, Recon2 was known to generate ATP without the import of any carbon substrate. This issue was addressed in Recon2.2 by introducing a new compartment to the model (the mitochondrial intermembrane space). The possible unaccounted influx of carbon substrates via sink reactions, which could also be responsible for this behavior, was also addressed by constraining the flux through these reactions to zero. All transport reactions between the cytoplasmic and external compartments of the model were constrained (i.e., upper and lower bound set to zero). Only the reactions relevant to the intracellular transport of glucose (Recon2.2 reaction id: GLCGLUT2, GLCt2), palmitate (reaction id: HDCAFAPMtc), oxygen (reaction id: O2t), carbon dioxyde (reaction id: CO2t), and water (reaction id: H2Ot) were left unconstrained.

To identify the fuel mix utilized by the models during the fast to fed transition, we computed the Respiratory Quotient (RQ) using the following relation:(1)$$RQ=CO_{2}^{out}/O_{2}^{in}$$

In fasting conditions, the plasma concentration of glucose is low, and skeletal muscle uses fatty acids as an energy substrate. After a meal, the plasma concentration of glucose rises, and insulin is secreted by the pancreas in response. Insulin signals to skeletal muscles and to other tissues to use glucose for energy production. The oxidation of fatty acids such as palmitate is inhibited, and fatty acids are instead stored in adipocytes as energy reserve. This is known as the fast to fed transition.

We reproduced this transition by progressively changing the upper bound of the glucose and palmitate exchange reactions. In the fasted condition, the maximal influx of glucose and palmitate was restricted to 0.5 mM/h and 0.38 mM/h, respectively. In the fed condition, the maximal influx of glucose and palmitate was restricted to 4.5 mM/h and 0.034 mM/h, respectively. All other exchanges, except water and protons, were deactivated. The LP solver chose a combination of palmitate and glucose from the medium to fulfill the cellular objective. By maximizing ATP consumption instead of ATP production, the models generated ATP using the optimal pathway, thus eliminating a potential source of bias. To improve the stability of the solutions, we decided to employ parsimonious flux balance analysis (PFBA) [18]. This extension of classical FBA identifies the flux distribution that fulfils the objective while minimizing the number of reactions that carry flux. By constraining the flux through the ATPH reaction, we simulated a condition of reduced energy expenditure. This allowed us to simulate the effect of reducing energy expenditure on fuel choice. (Figure 1B). The analyses were performed in Python 3.7 using the Cobrapy package (v. 0.22) [28]. The Gurobi solver (v. 9.1) was used to perform flux balance analysis.

## 5. Conclusions

A new description of the fast to fed transition enables the investigation of the interactions between energy expenditure and fuel choice using constraint-based models. Even though it is limited to the analysis of glucose and palmitate metabolism, our model is rich enough to describe metabolic flexibility. Recon2.2 and Recon3D were improved by accounting for unbalanced protons in glucose transport reactions. Simulating low energy expenditure conditions reproduced phenotypes linked to metabolic inflexibility in several human metabolic reconstructions. Predicted RQ changes are independent of substrate intake fluxes. Patient-derived models of skeletal muscle metabolism can capture the metabolic adaptations following a resistance training intervention and can be used to investigate the variability in the individual responses to metabolic interventions. Physical activity can restore metabolic flexibility.

## Figures and Tables

**Figure 1 metabolites-11-00695-f001:**
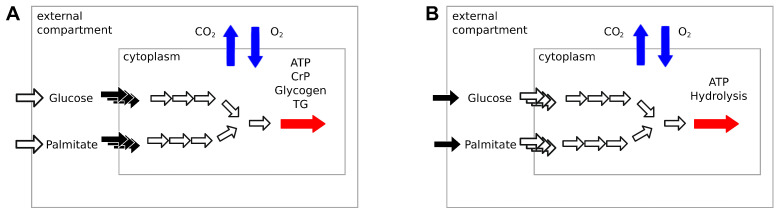
Simplified representation of two different descriptions of the fast to fed transition in a constraint-based metabolic model. (**A**) Architecture of the simulation presented in [11]: change in nutrient availability during the fast to fed transition is modeled by modulating the flux through glucose and palmitate transporters, the reactions transporting substrates between the external and the cytoplasm compartments (black arrows). Production of ATP, creatinine phosphate (CrP), glycogen and triglycerides (TG) was used as objective reaction (red arrow). The availability of glucose and palmitate in the external compartment is assumed to be infinite. (**B**) Architecture of the simulation presented in this study. The fast to fed transition is modeled by modulating the amount of nutrients available in the external compartment through exchange reactions (black arrows). ATP phosphodiester bond hydrolysis (ATPH) is used as objective function (red arrow). The models are free to choose the optimal mix of substrates to optimize the flux through the objective function. RQ is defined as the ratio between CO${}_{2}$ efflux and O${}_{2}$ influx (blue arrows) in both implementations. Blank arrows represent reactions that were left unbounded.

**Figure 2 metabolites-11-00695-f002:**
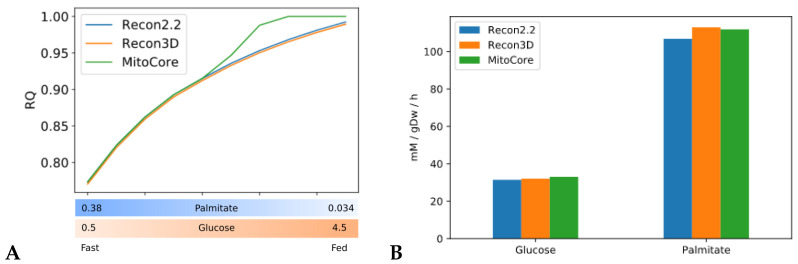
Simulations of fast to fed transition highlight heterogeneity of model predictions. Validation of our of the fast-to fed transition. (**A**) RQ values predicted by three human constraint-based models (Recon2.2, Recon3D and MitoCore) during the fast to fed transition with the objective function ATPH left unconstrained (upper bound ATPH = 1000 mM/gDw/h). X axis: upper bound values for palmitate and glucose exchange reactions during the fast to fed transition (in mM/gDw/h). (**B**) ATP yields for glucose and palmitate across the three models.

**Figure 3 metabolites-11-00695-f003:**
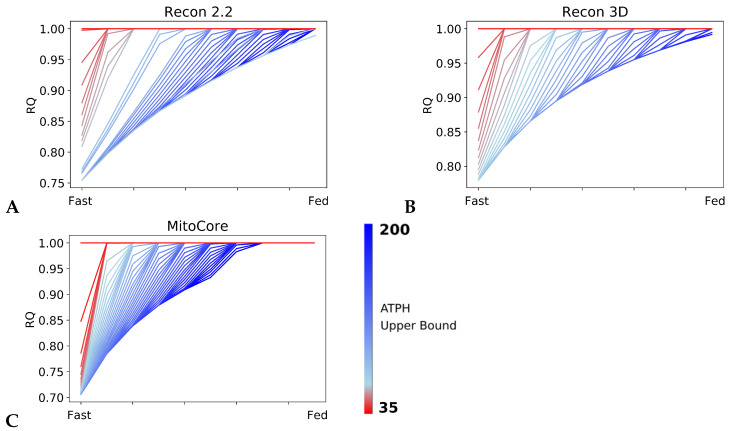
RQ is sensitive to changes in ATP hydrolysis rate. RQ values during the fast to fed transition simulated for different rates of ATP hydrolysis in Recon2.2 (**A**), Recon3D (**B**) and MitoCore (**C**). The upper bound of the objective reaction (ATPH) was progressively decreased from 200 mM/gDw/h (blue line) to 35 mM/gDw/h (red line). In all models, as ATP hydrolysis rate decreases, RQ values approaches a constant value (RQ = 1.0) faster during the fast to fed transition.

**Figure 4 metabolites-11-00695-f004:**
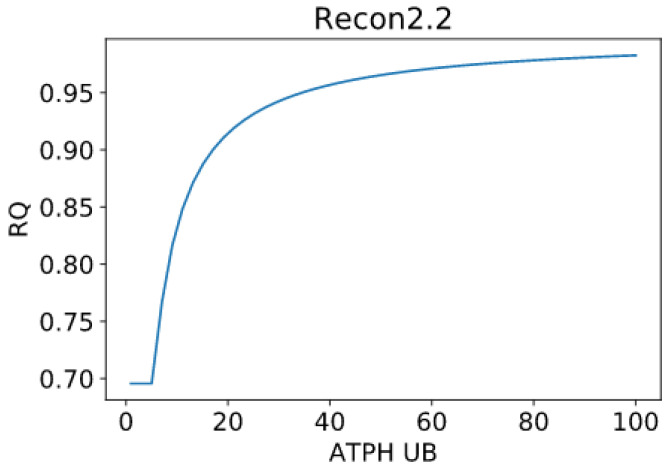
RQ changes are independent on intake fluxes. Predicted RQ for varying levels of ATPH upper bound. Intake bounds for palmitate and glucose were kept constant. Glucose uptake upper bound: 4.5 mmol/gDw/h. Palmitate uptake upper bound: 0.38 mmol/gDw/h.

**Figure 5 metabolites-11-00695-f005:**
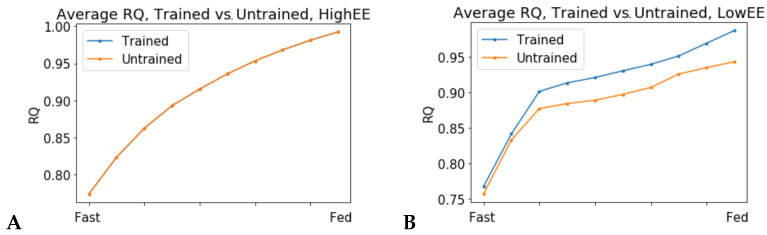
Simulations in low energy expenditure conditions show heterogeneity of individualized models’ predictions. RQ values predicted by a set of 24 patient-derived models of skeletal muscle metabolism. (**A**) High energy expenditure conditions (ATPH bound = 1000 mM/gDw/h). All models predict the same RQ values during the fast to fed transition and have overlapping RQ profiles. (**B**) Comparison of trained vs. untrained subgroups. Low energy expenditure condition (ATPH upper bound = 35 mM/gDw/h). In this condition, untrained models predicted lower RQ values on average and low variability between the fast and fed conditions than trained models. These two phenotypes are associated with metabolic inflexibility.

**Figure 6 metabolites-11-00695-f006:**
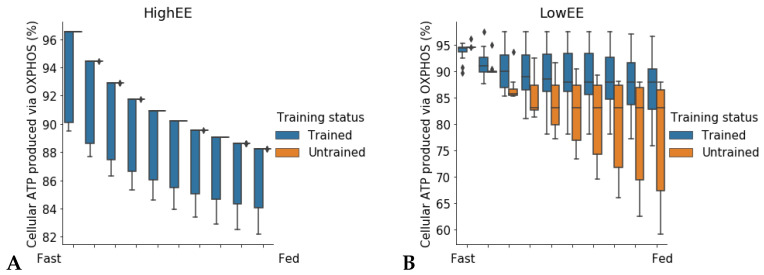
Increased utilization of oxidative phosphorylation (OXPHOS) in trained models in response to low energy demands. Percentage of total cellular ATP produced was measured as flux through the adenine nucleotide translocator (ANT) reaction (reaction ID: ATPtm). Twenty-one models were included in the analysis (N trained = 12, N untrained = 9). (**A**) High energy expenditure. (**B**) Low energy expenditure. In low EE conditions, trained models produce a higher percentage of total ATP from OXPHOS than untrained ones. Untrained models show a larger variance in the percentage of total ATP obtained from OXPHOS than untrained models.

## Data Availability

The gene expression data presented in this study are openly available in GEO, the Gene Expression Omnibus at accession number GSE28422.

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
