# Peer review of "Simulating Metabolic Flexibility in Low Energy Expenditure Conditions Using Genome-Scale Metabolic Models"

_metabolites, 2021, doi:10.3390/metabo11100695_

Round 1

Reviewer 1 Report

Metabolic flexibility refers to the capacity to modulate the
level of daily fuel oxidation to alteration in fuel availability. Studies have reported  that physical inactivity  is one of the primary causes in the development of metabolic inflexibility. The authors in this manuscript investigated the link between physical inactivity and metabolic inflexibility which is not novel. The authors in this study reported that the reduced ATP consumption due to lack of physical activity led to metabolic inflexibility. However, the authors didn't determine the ATP consumption threshold below which metabolic inflexibility would develop. Determining the threshold level would be beneficial in order to refine the physical activity recommendations in the general population.

Reviewer 2 Report

In this article, the authors studied metabolic flexibility of human organism which changes based on nutrient availability and energy requirements. The authors studied metabolic flexibility for fat and carbohydrate metabolism. In humans, transitions between fat and carbohydrate metabolism occurs due to interplay between insulin/glucagon hormones with insulin signalling switch to carbohydrate usage and glucagon signalling switch to usage of fatty acids such as palmitate.

The authors measured different patient’s RQ which were ranging from 0.7 and 1. When for each patient’s condition, they created patient specific genome-scale models. For example, if RQ was 0.7 (pure fat metabolism), they constrained palmitate uptake rate up to 0.38 mM/h (high fatty acid uptake) and glucose uptake rate up to 0.5 mM/h (low carbohydrate uptake). If RQ was 1.0, they constrained palmitate uptake rate up to 4.5 mM/h (low fatty acid uptake) and glucose uptake rate up to 0.034 mM/h (high carbohydrate uptake). Although the researchers only focused on interplay between carbohydrate and oxygen metabolism as all other uptakes of metabolites were switched off, the strategy used in this paper could be one of the standard strategies used for descriptive purposes of genome-scale models. As a conclusion, the manuscript can be accepted for publication as is.

Reviewer 3 Report

Genome-scale metabolic models (GEMs) have been employed to simulate metabolic flexibility by computing the Respiratory Quotient (RQ) in different nutritional conditions. Fewer studies have examined the role of physical activity on metabolic flexibility and GEM studies did not consider energy expenditure. The authors propose a description of fast to fed transition that allows to simulate various levels of physical activity affecting the fuel choice in GEMs. They apply ATPH as an objective function. The main difference of their approach to the alternative approaches is that they constrain glucose and palmitate through exchange reactions to simulate the fast to fed transition. They demonstrate their approach on a set of GEMs including two versions of Recon, MitoCore and a set of patient-derived models of skeletal muscle metabolism.

Comments:

Line 153: These models were developed... Please provide more details on how the data were acquired and how these models were developed (i.e., please provide a brief description in section 4). 

Please indicate more clearly what kind of methodology was used to assess the metabolic fluxes and why. At the end you state that the Gurobi solver was used to perform FBA. FBA has several limitations and does not provide a unique solution. Why did you decide to use FBA?

Line 159: Figure 4 instead of Fig4.

Currently, some of the figures are placed immediately after the subsection titles. Figures should only appear after their reference.

Reviewer 4 Report

I have a number of concerns about the results and their interpretation (or lack of it) in this paper, and as a consequence I lack confidence in the validity of the manuscript.

1. Firstly, the authors criticise a previous attempt (ref. 11) to study RQ of muscle under varying inputs of glucose and palmitate by stating that using fixed input rates for these externals predetermined the resulting RQ from the model simulations. I would not defend the objective function used by those authors, but I think the claim that the output was predetermined is unfair. Irrespective of the set of model reactions and their fluxes in a solution to any GSM, in a properly mass-conserving GSM, the steady state must be expressible as a balanced stoichiometric equation in the external metabolites. In the model of reference 11, the principal elements of this would be glucose + palmitate + O2 → CO2 + glycogen + TAG. Since glycogen and TAG were variables of the model, there is scope for flexibility in the relative usage of glucose and palmitate for energy production. I will return below to this issue of an overall stoichiometric equation for the steady state solution.

2. The profiles for the results from Recon2.2 and Recon3D in figs. S2 and S3 are unusual, if not pathological. It is not sufficient to say (p.5) “Certain behaviours cannot be explained…”. When scanning constraints in an FBA model, the flux responses are piecewise linear (because it’s linear programming). Changes of slope correspond to points where the set of reactions in the solution changes, so the metabolic alteration driving this can easily be found by comparing the reactions gained or lost each side of the inflection point. The reasons for the observed behaviour need to be investigated.

3. Of possible relevance to this are known shortcomings of the Recon models. The involvement of FAD in beta oxidation and the electron transport chain is not correctly represented in these models (and many others besides). Appropriate corrections were made by Bakker’s group (BBA 1865, 360-370, 2019). In addition, one of the authors of ref. 14 (Recon2.2) admitted to me that the model could generate ATP without the import of any carbon substrate, and it should be checked that this is not true of the versions used here, as it would obviously have a confounding influence.

4. There are indications that mass conservation in the models might need to be checked. The RQ value of 1.015 shown in Appendix A cannot simply be attributed to reaction CYOOm3i alone, since that reaction is mass-balanced for oxygen, which could imply the existence of an unbalanced reaction elsewhere in metabolism linked to this reaction. Again, what are the overall stoichiometric equations for the steady state solutions with an RQ of 1.015? Are they atomically balanced? Figure S1 for the Recon2.2-based individualised models shows a number of results of RQs below 0.7. How can these arise from catabolism of palmitate and glucose? What are the stoichiometric equations for the steady states in these cases?

Reviewer 5 Report

The study uses previously published human metabolic models to examine RQ values as a function of glucose and fatty acid metabolism.  The authors change the boundary conditions (permitted flux of glucose and fatty acids) and record the resulting RQ values.  The same approach is applied to published human models that use patient transcript data to remove reactions from the model.

The results do not appear to be novel or overly impactful. The authors change the permitted flux of glucose and fatty acid which necessarily results in simulation results indicating a shift from one substrate to the next. As written, it seems the results are a direct result of the user defined boundary conditions and not any new metabolic insight.  The work needs to better describe what is novel and demonstrate that the results are not a result of, what appears to be, rather obvious results of user defined boundary conditions.

I agree with the authors that many FBA results are predetermined by the constraints the modelers apply.  Many studies are simply data fitting exercises that bias results by carefully selecting constraints and boundary fluxes.  Unfortunately, the current work seems to do the same things.

The authors state previous studies did not account for energy expenditure but then go on to state the previous studies maximized ATP etc as a function of substrate fluxes.  It seems the previous studies did examine energy expenditure. I don’t feel the authors treatment of glucose and palmitate transport is notably better or worse than the previous studies and therefore also has bias.

Lines 66-67: The authors need to be careful with the ATP consumption and synthesis statements.  I believe they are referring to the production and hydrolysis of phosphodiester bonds not the actual ATP metabolite. Maximizing the production of phosphodiester bonds or the hydrolysis of phosphodiester bonds should provide the same results using an FBA model because of the enforced steady state.  The authors’ arguments that their current approach is better than previous studies is not convincing unless, the previous study did in fact examine the de novo synthesis of ATP molecules and not specifically the generation of phosphodiester bonds.

Line 73: how are any of the predictions validated? The different models give different results and the predictions are simply predictions.  I do not feel anything is validated.  Line 94+, 121-124: the RQ value is simply a projection of an electron balance, each glucose carbon carries 4 oxidizable electrons and each fatty acid carbon carries 6 oxidizable electrons and each O2 molecule can acquire 4 electrons, if the model returned RQ values outside of the 0.7-1 range  – it would mean the model is poorly built and violates mass balances/first law of thermo (the authors state no reduced byproducts are permitted), having RQ values between 0.7- and 1 only shows the model  reactions are mass and electron balanced  and therefore do not create nor destroy mass, this does not validate the proposed physical inactivity hypothesis in any way

The paper over emphasizes the authors’ interpretation of the results.  The FBA model does not account for physical inactivity, it is the authors’ applied constraints and maximization criteria that are proposed to reflect a scenario  relevant to physical inactivity.

Line 89: the units in this sentence are not consistent, the document lists a volumetric rate (mol/L/h) for the substrates but the ATP hydrolysis rates are given on a specific rate basis (mol/ g cdw/h). How are the volumetric rates translated into specific rates?

Line 89-91: the authors constrain the model thus biasing the results with the substrate boundary conditions, the introduction/abstract fault previous work for doing the same thing.

Line 112+: the different constraints and different model parameters result in slight differences between the three models, which model is the best, which simulation is the most accurate?  How do these differing results contribute to the claim the results are validated?

Figure 3: The term ‘ATP turnover rate’ implies enzyme kinetic parameters, ie enzyme turn over rate which is not included in the model, perhaps use ATP hydrolysis rate

Line 125+: the models preferred glucose during the fed conditions because the authors increased the permitted glucose flux and decreased the permitted fatty acid flux (lines 89-91) for that condition, are these results really a surprise or are details missing so I don’t appreciate some aspect of the analysis? are the simulations applying additional constraints like the parsimonious constraint?

Line 145: it is argued the results are an intrinsic property of the network, this is not true, the results are an intrinsic result of the constraints the modelers put on the simulations

Line 146: it is not clear from the document what the skeletal muscle models entail making interpretation of the results challenging, how were the mRNA data mapped to the models? Were the model fluxes constrained proportional to the mRNA or was it a binary on/off? Predicting RQ values outside of 0.7-1 would suggest the models were compromised and not mass balanced (or the model permitted secretion of lactate), if the flux through oxphos was low then the model must have predicted the production of lactic acid as otherwise the mass/electron balance would have been violated

The results appear to be the direct outcome of the constraints the authors placed on the simulations.  It is not clear if the authors appreciate the mathematical basis of the simulations which may lead to claims that exceed the capabilities of a stoichiometric model.  The model is simply a stoichiometric matrix, arguments about physical activity etc are the user applying constraints/bias.

Author Response

Kind regards,

The authors.

Round 2

Reviewer 3 Report

The authors have adequately addressed my comments.

Reviewer 4 Report

The authors have accepted that there were flaws in their previous analysis and that the majority of my diagnosis was essentially correct.  They have fixed the problems and re-run their analyses, so I now have confidence in their results.

A useful by-product of this is that they have documented the issues that arose from applying these published models, so the paper now also provides examples of the types of checks that need to be made by other scientists who seek to use them.